# RSP4J: Towards an unifying API for RDF Stream Processing

Pieter Bonte[1], Riccardo Tommasini[2], Emanuele Della Valle[3] and
Femke Ongenae[1]

1 Ghent University - imec, Ghent, Belgium
`firstname.lastname@ugent.be`
2 University of Tartu, Data System Group, Estonia
`riccardo.tommasini@ut.ee`
3 Politecnico di Milano, DEIB, Milan, Italy
`emanuele.dellavalle@polimi.it`

**Abstract.** Within the RDF Stream Processing (RSP) community, several languages and engines have been proposed for continuous querying over RDF streams. However, they lack standardization and shared design principles, making comparative research and benchmarking extremely hard. This has led to the design of RSPQL, i.e., a unifying model for RSP. However, an RSP API for the development under RSPQL semantics was still missing. We propose RSP4J, a flexible API for the development of RSP engines and applications under RSPQL semantics. RSP4J will be presented at the ESWC 2021 Resources Track, while this paper also identifies the future plans for RSP4J.

## 1 Introduction

Data-intensive domains, such as the Internet of Things and social media have unveiled the streaming nature of information [6]. Stream Reasoning (SR) and in particular RDF Stream Processing (RSP) is the research area that combines Stream Processing and Semantic Web technologies in order to extract actionable insights from heterogeneous data streams [9].

The SR community has proposed various RSP languages as extensions of SPARQL that support some form of continuous semantics, e.g., C-SPARQL, CQELS-QL, SPARQL$_{stream}$, and Strider-QL. Each of the languages is typically paired with a prototype engine to help prove the feasibility of the approach and study the efficiency. However, the lack of standardization and shared design principles of these RSP engines are obstructing the growth of the community. As prototyping efforts remain isolated, the costs of development and maintenance of prototypes remain on the shoulder of individuals researchers. Furthermore, each of these engines apply different *execution semantics*, making comparative research on and benchmarking of RSP engines extremely hard, as the semantics of different RSP languages do not completely overlap and no clear winner can be identified [7].

In order to solve this issue, the community proposed RSPQL [8], a reference model that unifies existing RSP dialects and the execution semantics of existing RSP engines. RSPQL is a first step towards a community standard, unfortunately, existing prototypes still do not follow shared design principles. However, *an API based on RSPQL would reduce the maintenance cost of existing engines, foster adoption of RSP engines, open new research opportunities in Stream Reasoning.*

We present RSP4J[1], a configurable API for building RSP engines under RSPQL semantics, presented at the ESWC 2021 Resources Track [11], and identify the future directions for RSP4J.

## 2   RSP4J

RSP4J aims to solve the following use cases:

- **Fast prototyping**: The effort of designing good prototypes is high, resulting in prototypes with a minimal set of requirements and shared design principles. RSP4J aims at providing the necessary abstractions to speed-up prototyping while providing the needed design principles. This allows to easily add new operators, define new types of data sources, or experiment with new optimization techniques without huge amounts of manual efforts.
- **Comparable research & benchmarking**: as current RSP engines do not share the same *execution semantics*, it is hard to reproduce the behavior in a comparable way. By building upon the semantics of RSPQL and providing the necessary abstractions, RSP4J aims at fair comparison, allowing to thoroughly benchmark the various RSP approaches and fairly compare their weaknesses and strengths.
- **Education**: existing prototypes do not allow to inspect the engine's behavior or investigate the various levels of abstractions. In order to simplify teaching, RSP4J provides the necessary abstractions to isolate and learn about the different components that make an RSP engine, while adhering to the underlying theoretical framework proposed by RSPQL.

In order to so, RSP4J provides all the necessary abstractions to develop RSP engines. RSP4J consists of five core modules, each providing the necessary abstractions to simplify the definition of RSP engines under RSPQL semantics. The various module are, as depicted in Figure 1:

a **Querying**: allows to write RSP programs in a declarative manner, based on the RSPQL syntax.
b **Streams**: the stream abstraction allows to provide custom implementation of a data stream. Two options are available, inspired by VoCaLS [10], i.e. the *Web Stream* represented as a Web resource and the *Data Stream* as a data source.

---

[1] https://github.com/streamreasoning/rsp4j

c **Operators**: RSP4J includes abstractions for the following RSPQL operators:
  - *StreamToRelation*: converts RDF Streams to finite RDF Data, e.g. through *Time-Based Sliding Windows.*
  - *RelationToRelation*: allows to perform SPARQL 1.1 algebraic expressions over a converted finit RDF Dataset.
  - *RelationToStream*: allows to go convert back from solutions mapping as a result of the SPARQL evaluation, to RDF Streams.

d **Streaming Data Set** (SDS): as specified by RSPQL, the SDS is an extension of the SPARQL dataset to support continuous semantics. It collaborates with the StreamToRelation operator to freeze and poll the active window content.

e **Engine & Execution semantics**: this module provides the abstractions to control and monitor the RSP engine its lifecycle.

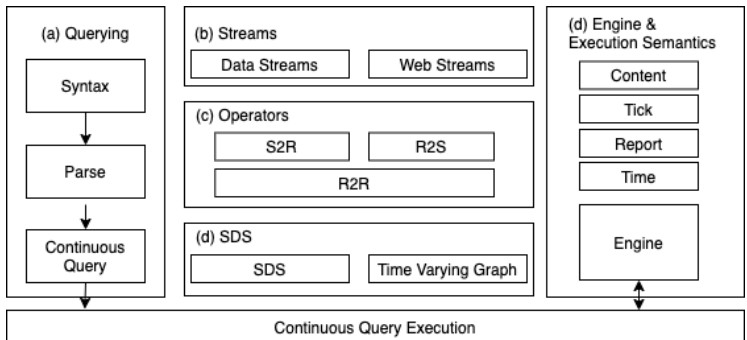

Fig. 1: RSP4J's Modules: (a) Querying (b) Streams, (c) Operators, (d) the SDS, and (e) Engine and Execution Semantics

## 3   The Future for RSP4J

In this section, we identify the planned initiatives for the future of RSP4J.

**Engine support:** Currently, two RSP engines are already building upon RSP4J: (i) YASPER[2], which is a strawman implementation based on Apache RDF Commons[2], and C-SPARQL 2.0[3] a new version of the C-SPARQL engine [1]. We plan to align even more engines with RSP4J, such as $Morph_{stream}$, CQELS and Strider.

**Modules Library:** We will provide a library with different implementations of the various RSP4J modules and components and documentation of their performance trade-offs, e.g. memory can be sacrificed in order to obtain increased

---

[2] https://github.com/streamreasoning/rsp4j/tree/master/yasper
[3] https://github.com/streamreasoning/csparql2

throughput. Due to the abstractions and the modularity of RSP4J's design, different implementations can easily be plugged-in. This allows to offer and evaluate performance trade-offs while fast prototyping. Similarly, we will provide the necessary components to plug-in different levels of reasoning expressivity, e.g. RDFS, the various OWL2 profiles, or ASP. Furthermore, we will provide existing optimization algorithms for the various API modules, e.g. C-Sprite [4] or LiteMat [5] as reasoning optimizations in the R2R operator.

**RSP Builder Framework:** In order to further speed-up prototyping, we will further abstract RSP4J, providing a builder framework that allows to select the needed modules from the *Modules Library* (or custom implementations) and built it into a customizable RSP engine. This allows to further simplify the API and improve comparable research.

**Language support:** At this point, RSP4J is fully Java-based, we aim to abstract the RSP4J specification and provide access to other languages, such as Python. This will further improve the adoption of RSP4J.

**Alignment with other Stream Reasoning Frameworks:** We will investigate how RSP4J can align with other efforts within the SR community, besides RSP, e.g. LARS [2] or Cascading Reasoning approaches [3].

**Acknowledgments:** Pieter Bonte is funded by a postdoctoral fellowship of Fonds Wetenschappelijk Onderzoek Vlaanderen (FWO) (1266521N).

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
