# OpenReview forum: "RSP4J: Towards an unifying API for RDF Stream Processing"
_eswc-conferences.org/ESWC/2021/Conference/Poster_and_Demo_Track — Submitted to ESWC2021 P&D_

### Official Review · ~Umutcan_Simsek1 · 2021-04-13
**nice work with potentially high impact, however not clear how much it differs from the already accepted resource paper**

**Rating:** 6
**Confidence:** 3

**Review:**

# content of the poster
the poster is about a proposed common API to reduce the heterogeneity of RDF Stream Processor implementation. It serves as a supplementary material to an accepted resource track paper.

# relevance to the Semantic Web, potential significance, topicality, clarity.
I think the poster is all fine from these fronts. It is clearly written and the proposed API is highly relevant to the semantic web field, particularly to the stream processing. Given the developments in IoT field and industrial applications of semantic technologies, it is a very timely work.

# originality
This is the main reason that I cannot recommend this paper with a high score. The work is overall I believe can have a high impact but it is somewhat a summary of an already accepted resource paper and it is hard to judge if this poster brings anything that is not already provided by the resource paper. Although there is no link to the accepted paper in the references, the version I found basically has every significant bit covered about the API (even with a larger detail). There is a very little new content regarding future directions, however this could be easily part of the future work part of the other paper, not sure if it justifies a standalone publication.  IMHO, it would have been better to submit a demo paper that demonstrates some concrete use cases and implementation details.

# summary
AFAIK, such supplementary papers are welcome at ESWC, so I will give a "6". However, my concern regarding the originality stands as I am not certain what level of new content is expected from such supplementary poster papers.

**Anonymity:**

No, I would like my review to be deanonymized.

---

### Official Review · AnonReviewer4 · 2021-04-14
**The paper is relevant although it may include duplicated content**

**Rating:** 6
**Confidence:** 3

**Review:**

The paper presents RSP4J, an API for RDF stream processing. Besides the last section ('The future for RSP4J'), it is not clear whether the content is a summary of the accepted resource paper or if it is original. Details about RSP4J could be omitted if it were to expand the last section, to include a thorough development plan.

For instance, the authors could present a landscape of stream processing engines that could align with RSP4J, with e.g. an estimate of the required alignment effort and whether that effort is going to be supplied by the developers of RSP4J or the authors of the engine.

The paper could also include an initial draft of what the RSP builder framework would look like.

Remarks on typesetting/references:
 - ref [7] is empty
 - on the Github repo, the link for RSP-QL in the README is broken
 - in the 'Engine support' paragraph, there is a (i) but no (ii)
 - 'the various module' -> 'the various modules'

**Anonymity:**

Yes, I would like my review to remain anonymous.

---

### Official Review · AnonReviewer1 · 2021-04-14
**Preliminary work**

**Rating:** 4
**Confidence:** 3

**Review:**

The paper introduces RSP4J, an API intended to unify applications using RSPQL semantics.

The motivation for the research is quite convincing as different standards make it hard to compare and evaluate different approaches. However, the tool's description of the modules as well as "what it aims to solve" is not very well described. From the text, it is not clear what has been done nor how it actually works. For instance, the biggest problem I have is the section where the modules follow after a series of aims of the tool, but it is not clear to me how these are connected to one another, nor how they relate to the future work in the following section.

Overall the paper reads as preliminary work that needs a bit of polishing - which could be alright for the poster track. However, since the paper's content is presented at the resource track of the conference, I would not recommend it for acceptance at the P&D track as well.

**Anonymity:**

Yes, I would like my review to remain anonymous.

---

### Official Review · AnonReviewer3 · 2021-04-16
**Unclear nature of the submission**

**Rating:** 5
**Confidence:** 4

**Review:**

The submission at hand is about an API for RDF stream processing, which is presented at the resource track of the conference.

Having read the submission, the nature of the submission is still unclear to me (poster/demo/vision/position/outrageous idea). Maybe the abstract can help: "RSP4J will be presented at the ESWC 2021 Resources Track, while this paper also identifies the future plans for RSP4J.". Correspondingly, I would have expected a vision paper with more visionary content. Also, it is unclear to me what the authors want to show using the submission at hand, ie. how you present an API. Will you show Javadoc?

When compared to the full paper, the submission is a bit of a misnomer: The submission at hand is "towards a unifying API", while the resource paper presents the API, so I wonder about the added value of the submission at hand, and the temporal ordering.

As this submission accompanies an accepted resource paper, I assume the criteria relevance to the Semantic Web, originality, potential significance, and topicality, as fulfilled.

My criticisms from above stand, but I want to encourage the discussion of the resource paper in the poster/demo track, hence for me the paper is marginally below threshold, ready for other reviewers to vote it above the threshold.

Minor comment:
The bibliography entry for reference 7 is empty

**Anonymity:**

Yes, I would like my review to remain anonymous.

---

### Official Review · Program_Chairs · 2021-04-18
**Metareview: Reject (Unclear additional contribution over Resource Track paper)**

**Rating:** 4
**Confidence:** 5

**Review:**

While the reviewers are generally quite positive about the work, they express concerns that it is not clear what the added benefit of this paper is beyond the resource track paper, finding various aspects of the poster paper to be unclear or in need of polishing. Given, in particular, the lack of clarity regarding the added value of the poster paper versus the resource paper (something that was required by the CfP), we have opted not to accept the paper.

**Anonymity:**

Yes, I would like my review to remain anonymous.

---

### Decision · Program_Chairs · 2021-04-19

Reject